# M-BioBERTa: Modular RoBERTa-based Model for Biobank-scale Unified Representations

## Abstract

Transformers provide a novel approach for unifying large-scale biobank data spread across different modalities and omic domains. We introduce M-BioBERTa, a modular architecture for multimodal data that offers a robust mechanism for managing missing information. We evaluate the model using genetic, demographic, laboratory, diagnostic and drug prescription data from the UK Biobank, focusing on multimorbidity and polypharmacy related to major depressive disorder. We investigate the harmonized and modular representations in M-BioBERTa for patient stratification. Furthermore, leveraging the learned representations to forecast future disease and drug burdens outperforms traditional machine learning approaches applied directly to the raw data.

## 1 Introduction

Integration of heterogeneous information has always been a vital focus in life sciences (Bertone & Gerstein, 2001). Especially the integration of omic data sets to explore common molecular factors behind multimorbidities (Rosenthal et al., 2023; Langenberg et al., 2023) using probabilistic graphical models (Moreau et al., 2003), statistical relational learning (Weiss et al., 2012), kernel methods (Lanckriet et al., 2004), and network propagation (Zhu et al., 2012; Cowen et al., 2017). The growing availability of electronic patient records (EHR), quantified self data, patient registries, and biobanks provides an unprecedented amount of heterogeneous data to learn latent patient representations, stratify patients and subtype diseases (Swan, 2013; Bycroft et al., 2018; Jensen et al., 2014; Siggaard et al., 2020; Landi et al., 2020; Yang et al., 2023; Placido et al., 2023), although the integration of time series data, such as the irregularly sampled drug prescription data (Allam et al., 2021; Chen, 2020; Xie et al., 2022) and multimodal data fusion with systematically missing modalities is still an open problem (Acosta et al., 2022; Steyaert et al., 2023; Moor et al., 2023).

Transformers have proven to be powerful candidates for multimodal data fusion and time series data analysis (Baltrušaitis et al., 2018; Xu et al., 2023; Nerella et al., 2023; Ahmed et al., 2023; Wen et al., 2022). We focus on three open issues in the multimodal application of transformers:

1. autonomous treatment of modalities,
2. mixed numeric-nominal data types,
3. non-uniformly sampled longitudinal data.

This article introduces a modular transformer model, M-BioBERTa, to integrate genetic, demographic, metabolomic, diagnostic, and drug prescription data with varying levels of availability in the UK Biobank (Bycroft et al., 2018). We demonstrate the representation learning ability of M-BioBERTa in patient stratification and prediction tasks related to the molecular background, multimorbidity, and the polypharmacy of major depressive disorder (Koné Pefoyo et al., 2015; Guthrie et al., 2015; Langenberg et al., 2023; Flint, 2023). Fig. 1 provides an overview of the data sets, the M-BioBERTa model architecture, and the evaluation workflow of our study. Following convention in the field, we use multimodal and cross-domain interchangeably.

## 2 Earlier works

The multimorbidity analysis of large-scale biobanks allows the identification of novel patient stratifications and shared molecular background of clusters of patients in the latent space of patient repre-

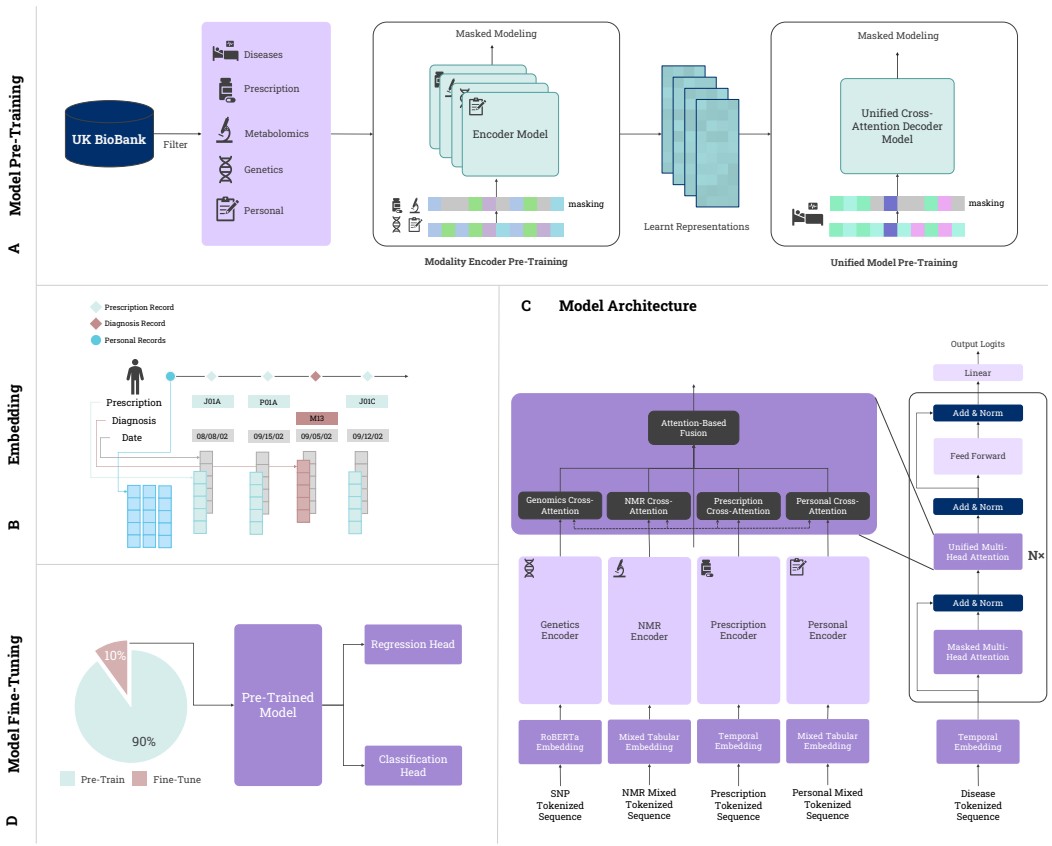

Figure 1: **Overview of data, methods, and architecture. (A) Training:** The model modular is not only architectural, but also manifest in the training phase. The encoder handling the different modalities of additional data, is pre-trained on datasets filtered on that modality. Then, the pre-trained encoders are used to train the unified model on the disease sequences. All models are trained to predict masked parts of their inputs. **(B) Embedding:** We introduce 2 new layers to facilitate the embedding of date information and tabular data. **(C) Architecture:** The RoBERTa decoder is augmented with new Unified Cross-Attention Layers, that processes input sequences from all modalities and merges them via an attention-based fusion mechanism. **(D) Fine-tuning:** To evaluate the effectiveness of the unified representation, we utilize 10% of our data to contrast the model's performance against conventional ML techniques.

sentations (Jensen et al., 2014; Prasad et al., 2022). Latent representations based on the metabolomic data from the UK Biobank have also been shown to increase performance in downstream predictive tasks (Buergel et al., 2022). In addition to non-longitudinal data, drug prescription time series data is an additional substantial resource, but its use is frequently hindered by heterogeneous coding, limited availability, and its longitudinal nature (Wu et al., 2019; Schwarz et al., 2022; Kiiskinen et al., 2023; Stroganov et al., 2022; Darke et al., 2022).

Transformers were rapidly adopted for multimodal and temporal biomedical data (Nerella et al., 2023). The BEHRT model could predict 301 conditions for future visits trained on the electronic health records of 1.6 million individuals (Li et al., 2020). Meng et al. (2021) proposed the Bidirectional Representation Learning model with a Transformer architecture on Multimodal EHR (BRLTM) to predict major depressive disorder (MDD) by integrating diagnoses, treatments, medications, demographic information, and clinical notes. The Med-BERT model demonstrated the advantages of structured inputs using ICD-9 and ICD-10 codes, transfer learning and fine-tuning over varying sample sizes between 100 to 50k (Rasmy et al., 2021). The Hi-BEHRT model contains a hierarchical extension to cope with long patient trajectories and increase transfer learning effects (Li et al., 2022). Yin et al. (2023) suggested structured multimodal learning to boost patient

stratification. The ExBEHRT model further extends the feature space through laboratory tests, integrates temporal side-information, and demonstrates the applicability of the model representations to identify subtypes of the disease and identify risk groups (Rupp et al., 2023). The Hybrid Value-Aware Transformer (HVAT) model facilitates joint learning from non-longitudinal and quantitative longitudinal data (Shao et al., 2023). Placido et al. (2023) compared the performance of deep learning methods using 6 million patients from the Danish National Patient Registry (DNPR) to predict the risk of pancreatic cancer from disease trajectories, in which transformers achieved the best AUROC values. Zhou et al. (2023) expanded the scope of multimodal learning to clinical images, texts, quantitative laboratory tests and structured information. For a critical evaluation of the applicability of transformers in time series prediction, see (Zeng et al., 2023), especially for the prediction of MDD (Zou et al., 2023). A standard EHR benchmark data set is the MIMIC data (Johnson et al., 2023); for an evaluation of large language models (LLMs) on the UK Biobank, see e.g., (Belyaeva et al., 2023). Finally, the Attention-based cRoss-MOdal fUsion with contRast (ARMOUR) architecture is designed to cope with systematically missing modalities (Liu et al., 2023), which shows similarities to our proposed model.

## 3    M-BioBERTa

In this section, we present M-BioBERTa, our novel architecture designed to handle large-scale biobank data. Unlike traditional models, M-BioBERTa uses several encoders to process different types of data modalities. This method allows our model to understand complex and systematically missing data more effectively and efficiently. First, we describe the basic structure of M-BioBERTa. This structure is key to our model's ability to handle different types of data efficiently. Next, we explain how our model uses multiple encoders. Each encoder adds its own piece to the overall picture, making our model more powerful in combining different types of modalities. An important part of M-BioBERTa is its decoder, which we discuss in detail later. The decoder uses an attention-based method to integrate information from the various encoders. This fusion is vital for our model to work well as it brings together all the different pieces of information in a meaningful way.

### 3.1    Transformers

The Transformer architecture represents a paradigm shift in sequence-to-sequence modeling, eschewing the recurrent layers common in prior models (Vaswani et al., 2017). Its foundation is primarily built on two mechanisms: self-attention and feed-forward neural networks.

**Input Embedding and Positional Encoding.** The model accepts an input sequence $X$, which is embedded in a continuous space using learned embeddings. A positional encoding supplements this to ensure that sequence-order information is retained.

**Self-Attention.** The self-attention mechanism, SA, allows the model to weigh the importance of different tokens in the sequence relative to each other. It is computed as:

$$\text{Attention}(Q, K, V) = \text{softmax}\left(\frac{QK^T}{\sqrt{d_{\text{model}}}}\right)V \tag{1}$$

**Multi-Head Attention.** Instead of a single set of attention weights, the Transformer uses multiple heads, MHA, to capture various aspects of the input information.

**Feed-Forward Neural Networks.** Each attention output is then passed through a feed-forward neural network FFNN, independently at each position.

**The Cross-Attention Mechanism.** In the cross-attention framework, the decoder attends to the encoder's output sequence. The process involves three main entities: the Query ($Q$) from the decoder, and the Key ($K$) and Value ($V$) from the encoder. The computation of the mechanism is the same as in Equation 1.

### 3.2    Temporal Embeddings

Within the landscape of medical data, longitudinal studies present unique challenges and opportunities, mainly due to the inherent temporal aspect of patient health trajectories. Continuous monitoring

and recording of patient health metrics, interventions, and results can reveal significant patterns in disease progression, therapeutic response, and external event-driven effects. In this section, we explain how we incorporate essential temporal information into the embedding layers of our model, allowing it to capture the complex temporal features of the medical field.

**Temporal Embeddings.** An architectural component tailored to transmute temporal information into continuous vector representations. Rather than using more complex structures or algorithms, this module adopts a traditional vocabulary-based embedding approach, translating temporal units into embeddings.

- **Year Embeddings.** Responsible for encapsulating longitudinal patterns and trajectories within medical data. Recognizing the importance of conditions and diseases that could develop or transform over several years, these embeddings are invaluable for prognostic tasks, such as forecasting disease trajectories or deciphering long-term treatment effects.
- **Month and Day Embeddings.** These facilitate the model's sensitivity to short-term temporal variations, such as cyclic medical phenomena or seasonal conditions. They ensure the model remains attuned to extensive and minute temporal changes.

The embedding creates a combined temporal representation by adding together the embeddings for each time unit: year, month, and day. This method ensures that each part of the time information adds its own unique information to the overall temporal representation. These temporal embeddings are added to the other embeddings within the model, integrating time-specific insights into the overall model architecture.

## 3.3 MIXED TABULAR EMBEDDINGS

This approach creates embeddings by combining token inputs, numerical values, token categories, positional and visit data. The tokenized part of the data utilizes an embedding layer, whereas the continuous data use a linear layer for the transformation. We employ a masking tensor to differentiate between the different types.

1. **Word Embeddings:**
   Given the input tensor of token IDs, $I$, and a masking tensor $M$ for continuous measurements:
   $$T_e = E_w((I \odot \sim M)_{int})$$
   Where $E_w$ represents the word embeddings matrix, and $T_e$ is the resulting token embeddings. $\odot$ denotes element-wise multiplication, and $\sim M$ is the logical negation of $M$.

2. **Measurement Embeddings:**
   Continuous measurements in the input tensor $I$ are converted to embeddings:
   $$M_e = E_m(I \odot M)$$
   Where $E_m$ is the linear layer that converts continuous measurements to embeddings of similar dimensionality, and $M_e$ is the resulting measurement embeddings.

3. **Combining Token and Measurement Embeddings:**
   The final embeddings from the token and measurement sources are obtained as:
   $$I_e = (\sim M \odot T_e) + (M \odot M_e)$$
   The comprehensive embedding for each token is the sum of its respective token or measurement embedding, positional embedding, token type embedding, and visit type embedding; the token type- $TT_e$ and visit embeddings $V_e$ are achieved with embedding layers as in standard architectures:
   $$E = I_e + P_e + TT_e + V_e$$
   Subsequent normalization and dropout operations are applied to $E$ for regularization.

The Mixed Tabular Embeddings provides a flexible framework for embedding heterogeneous data types commonly found in tabular datasets. This module captures a diverse range of information in its embeddings by considering tokens, continuous measurements, positional information, token types, and visit types.

### 3.4 UNIFIED CROSS-ATTENTION DECODER

The Unified Cross-Attention Decoder (UCAD) is designed to effectively integrate a variety of medical data types, such as patient history, drug interactions, genomic information, and laboratory results. Its core function is to create a cohesive understanding from these diverse data sources.

- **Dedicated Cross-Attention Units.** UCAD employs specialized cross-attention mechanisms, each focusing on a different type of medical data. These units are adept at identifying and aligning relationships between different data sources, ensuring that each modality contributes meaningfully to the overall understanding.

- **Attention-Based Fusion** This approach uses attention mechanisms to assess and weigh the relative importance of each data modality. By dynamically adjusting how different types of data are integrated, UCAD ensures that the most contextually relevant information is prioritized in the analysis.

**Attention-Based Fusion.** For a given input tensor $X$ with shape $[num\_sequences, batch\_size, seq\_len, hidden\_dim]$, and an attention weight vector $w$ of dimension $[hidden\_dim, 1]$:

1. **Attention Scores.** The scores are calculated by multiplying $X$ with the weight vector $w$:

$$s = X \cdot w$$

yielding a tensor $s$ reshaped to $[num\_sequences, batch\_size, seq\_len]$.

2. **Attention Probabilities.** Probabilities $p$ are obtained by applying the softmax operation over the $num\_sequences$ dimension:

$$p = \text{softmax}(s)$$

3. **Weighted Sum.** The final representation $O$ is a weighted summation of the input sequences based on their attention probabilities:

$$O = \sum_{i=1}^{num\_sequences} p_i \odot X_i$$

where $\odot$ denotes element-wise multiplication, aggregated across the $num\_sequences$ dimension.

The Attention-Based Fusion (ABF) approach assigns weights to each sequence in $X$ based on its content and the attention weight vector. This enables the combination of the sequences into a single representation, which captures the most important information from the set of sequences.

## 4 EXPERIMENTS

### 4.1 DATA

We used data from the UK Biobank (UKB), including basic descriptors and lifestyle factors (termed 'personal'), onsets of diseases ('diseases'), NMR metabolomic profile ('metabolomics'), an MDD-related genetic profile ('genetics'), and drug prescription time series data ('medications'). Table 4.1 shows the number of variables and samples for each block, and the supplementary data contains the list of the variables for each block. We used 1127 three-character coded ICD-10 disease categories. We extracted 700 measured SNPs for genes with confirmed relation to MDD by the DisGeNET platform (Piñero et al., 2020). We used all available NMR metabolic markers. We collected drug prescription data from the primary care records of approximately 165,000 participants in the UK Biobank. The data was sourced from the SystmOne practice management system, provided by TPP, which serves as the largest homogeneous data provider within the Biobank. Within this cohort, prescription data spans from 1990 to 2016 and is coded based on the British National Formulary (BNF) system. Each prescription entry includes the date, the BNF drug code and, when available, the drug name and quantity. We converted these BNF drug codes to the Anatomical Therapeutic Chemical (ATC) Classification System for our analytical purposes. This resulted in 176 ATC Level 3 classes used in the present study.

| Data Block | No. of Variables | No. of Samples |
|---|---|---|
| Personal | 178 | 502 504 |
| Diseases (ICD-10) | 1127 | 502 504 |
| Genetics (SNPs) | 700 | 488 377 |
| Metabolomics (NMR) | 249 | 121 724 |
| Medications (ATC) | 176 | 158 154 |

Table 1: **Overview of Data Used from the UK Biobank.** The samples of the different modalities are missing systematically.

## 4.2 MODELS

In our study, we compared different versions of M-BioBERTa, using the standard RoBERTa encoder as the transformer baseline. The following describes each version:

1. **RoBERTa Encoder (RoBERTa):** Used as a baseline model, the RoBERTa Encoder is trained exclusively on disease data, utilizing all available samples.

2. **M-BioBERTa:** This model incorporates all types of data, personal, diseases, metabolomics, genetics and medications, using every available sample.

3. **Modality-Specific Encoder:** These modules are tailored to specific data types and are trained only on the non-missing samples from their respective modalities.

4. **Pre-trained M-BioBERTa (pM-BioBERTa):** This variant begins with pre-trained Modality-Specific Encoder modules and is further trained on the entire dataset.

5. **Frozen Pretrained M-BioBERTa (fpM-BioBERTa):** In this model, the weights of the pre-trained modules are frozen, maintaining a balance between the initial training phase and subsequent refinements.

Our aim was to evaluate the performance of these models in processing and analyzing biobank data, focusing on how effectively each variant manages and interprets information from diverse data types. For further investigation into how key components effect performance, please refer to the Ablations section in Appendix A.4.

| Model | ICD Code Accuracy | ICD Block Accuracy | ICD Chapter Accuracy |
|---|---|---|---|
| RoBERTa | 20.06% | 25.47% | 36.15% |
| M-BioBERTa | 23.17% | 28.53% | 39.72% |
| pM-BioBERTa | 24.88% | 30.36% | 41.72% |
| fpM-BioBERTa | **25.42%** | **30.77%** | **42.25%** |

Table 2: **Model Performance Comparison Based on ICD Accuracy.** These metrics were used, to measure how well the model was able to perform that Masked Sequence Modelling tasks.

To evaluate the predictive performance (downstream tasks), we compare M-BioBERTa against standard machine learning models, such as linear regression (LiR), logistic regression (LR), multilayer perception (MLP), decision tree (DT), random forest (RF), and extreme gradient boosting (XGB). These models were trained on a tabulated version of the dataset A.6.

## 4.3 EVALUATIONS

The main goal of our evaluation was to investigate the subtypes of Major Depressive Disorder (MDD). We focus on two key aspects: multimorbidity (MM) and polypharmacy (PP). Multimorbidity refers to the presence of multiple diseases simultaneously, while polypharmacy deals with the concurrent use of multiple medications. We use the MM and PP scores to measure the burden of diseases and drugs, respectively. In our qualitative analysis of latent spaces, we looked at the first occurring class of diseases, the most frequent class of diseases, the multimorbidity scores, and

the polypharmacy scores. The modular design of M-BioBERTa allowed us to visualize the latent representations for each component.

Additionally, we evaluated how well our models could predict future medical conditions and MM/PP scores. We used data up to 2010 for training and predicted outcomes in the next 1 or 5 years. Given the low occurrence rates of some conditions, we chose the Area Under the Precision-Recall Curve (AUPRC) as our primary evaluation metric. For tasks related to predicting numerical values, we used the Mean Squared Error (MSE) metric.

We also experimented with various data-filtering approaches to enhance homogeneity. One such method involved segregating the data set based on birth year, grouping individuals into uniform cohorts. For example, we considered creating cohorts based on specific time spans, such as decades, to see if this would lead to more homogeneous groups. However, our experiments revealed a significant trade-off. Although filtering based on birth year and creating uniform cohorts did bring some level of homogeneity to the data, it also substantially reduced the dataset size.

*Data Partitioning for Model Training:*

- The UK Biobank (UKB) dataset was divided into separate parts for pre-training and fine-tuning, ensuring that the model was tested on new, unseen data.

- For the fine-tuning phase, we set aside specific portions of the data set for training and evaluation, maintaining a diverse and representative distribution of the data.

## 5 RESULTS

### 5.1 LATENT REPRESENTATION

The initial phase of pre-training the M-BioBERTa employed Masked Language Modeling (MLM, see A.3). Through this process, the model learns to generate embeddings in its latent space that are rich in contextual information and semantic associations, serving as an ideal foundation for downstream tasks. Given the multi-encoder architecture of M-BioBERTa, our model presents a unique setting. In addition to the fused latent space, a specific latent space corresponds to each encoder.

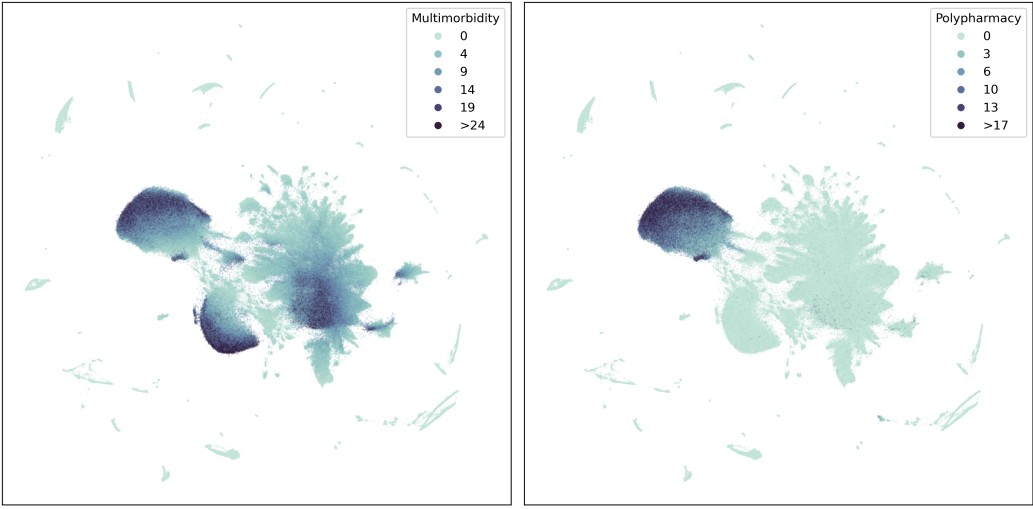

Figure 2: **The unified latent representation, colored by multimorbidity (left) and polypharmacy (right).** The figures show a 2D UMAP projection of the 312 dimensional latent space of the unified model. The structure in the coloring shows, that the model is able to place similar samples close together in the latent space. We provide a wider selection of latent space plots in the appendix B.

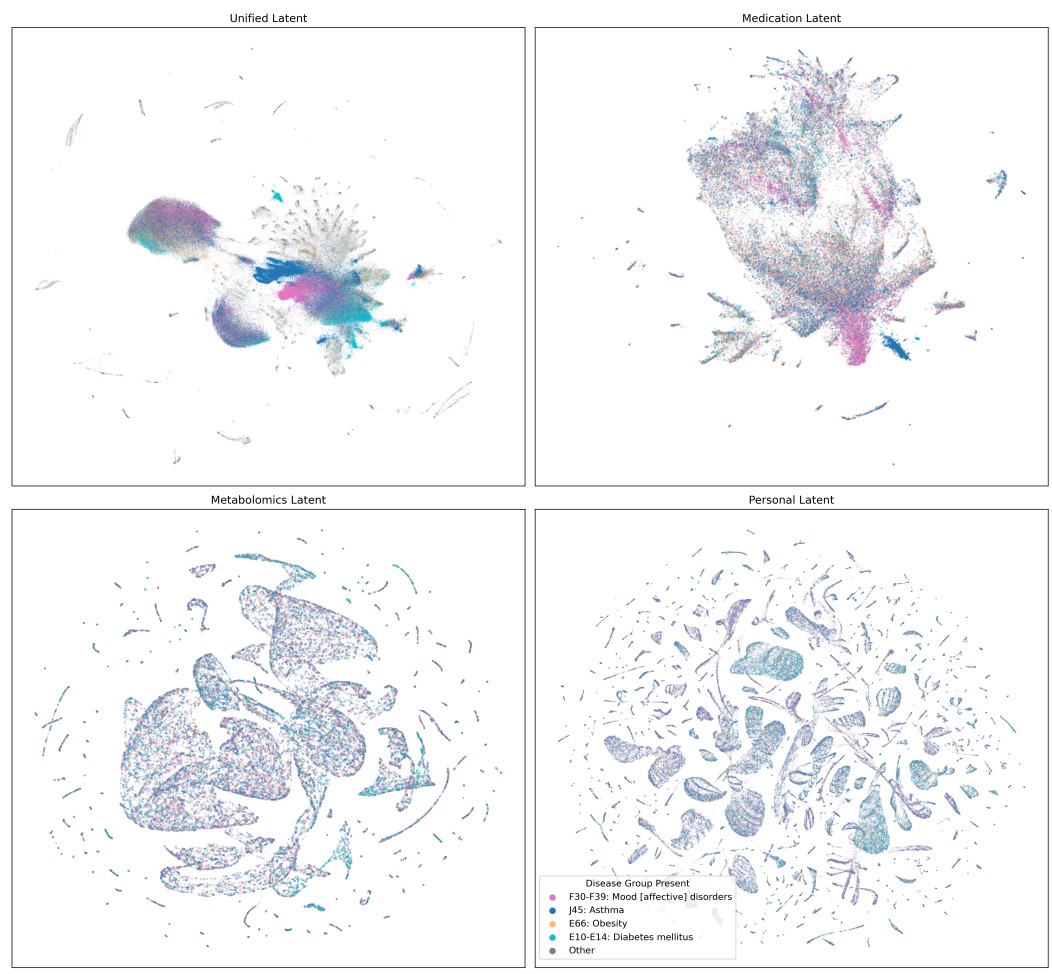

Figure 3: **Latent Representations of Pre-trained Models.** Each plot's latent space is color-coded based on the comorbidities associated with Major Depressive Disorder and showcases the samples used for training that specific modality. The disease groups are denoted by ICD10 codes. Note: The encoder pre-trained on genomics data is not depicted here and can be found in the appendix (see B).

## 5.2 FINE-TUNE

| Regression Datasets | | | Classification Datasets | | |
|---|---|---|---|---|---|
| **Metric** | **Mean** | **Std** | **Dataset** | **Pos Samples** | **Pos Percentage** |
| Disease Burden | 0.429 | 1.05 | F30-F39 | 787 | 1.62 |
| Unique Drug Burden | 0.668 | 1.79 | E10-E14 | 1237 | 2.55 |
| Drug Burden | 2.44 | 7.55 | I10-I15 | 4244 | 8.76 |

Table 3: **Statistics of Datasets for fine-tuning.** These sample were not used for pre-training the M-BioBERTa. All fine-tune datasets contain 48452 samples.

**Binary Classification.** We structured our data with specific temporal boundaries to ensure balanced representation and prevent temporal information leakage. We included data up to December 31, 2009, and our subsequent predictions spanned January 1, 2010, through December 31, 2014. A 5-year prediction period was selected to accumulate a statistically significant number of positive instances, ensuring a robust evaluation of binary classification tasks as in Table 5.2.

| Model | F30-F39 | | E10-E14 | | I10-I15 | |
|---|---|---|---|---|---|---|
| | AUROC | AUPRC | AUROC | AUPRC | AUROC | AUPRC |
| DT | 0.653 | 0.027 | 0.798 | 0.127 | 0.667 | 0.140 |
| LR | 0.658 | 0.028 | 0.801 | 0.147 | 0.657 | 0.138 |
| MLP | 0.666 | 0.044 | 0.782 | 0.114 | 0.681 | 0.145 |
| RF | 0.630 | 0.042 | 0.791 | 0.101 | 0.670 | 0.148 |
| XGB | 0.655 | 0.038 | **0.859** | 0.162 | 0.687 | 0.152 |
| RoBERTa | 0.713 | **0.276** | 0.756 | 0.394 | 0.660 | 0.303 |
| fpM-BioBERTa | **0.742** | 0.269 | 0.807 | **0.472** | **0.720** | **0.369** |

Table 4: **Performance metrics for classification tasks.** The models predict whether a patient will receive a disease from a group of diseases within 5 years of the data cut-off date. For tables showing all metrics, see AppendixA.7.

**Regression.** The same concept was applied to the regression data. However, the predictions were restricted to a narrower scope for regression tasks, specifically from 1 January 2010 to 31 December 2010. This one-year window was deemed optimal for discerning the model's ability to project continuous outcomes over a relatively shorter time.

| Model | Disease Burden | | Drug Burden | | Unique Drug Burden | |
|---|---|---|---|---|---|---|
| | MSE | MAPE | MSE | MAPE | MSE | MAPE |
| DT | 1.10 | 96.1 | 52.5 | 99.3 | 3.21 | 98.0 |
| LiR | 1.11 | 98.5 | 53.7 | 109.6 | 3.32 | 102.3 |
| MLP | 1.08 | 89.4 | 52.4 | 105.0 | 3.22 | 96.7 |
| RF | 1.08 | 95.8 | 52.1 | 99.0 | 3.2 | 98.5 |
| XGB | 1.08 | 95.6 | 52 | 100.0 | 3.2 | 98.5 |
| RoBERTa | **1.01** | 89.6 | 46.6 | 96.2 | 2.18 | 80.2 |
| fpM-BioBERTa | **1.01** | **80.7** | **23.8** | **51.8** | **1.15** | **44.6** |

Table 5: **Performance metrics for regression tasks.** The models are predict the number of diseases (Disease Burden), prescriptions (Drug Burden) and unique prescriptions (Unique Drug Burden) a patient will receive within 1 year of the data cut-off date. The Mean Absolute Percentage Error (MAPE) columns are given as percentages. For tables showing all metrics, refer to AppendixA.7.

## 6 CONCLUSION

Biobanks offer an unprecedented repertoire of multimodal data, including both non-longitudinal and longitudinal data; however, modalities are frequently systematically missing, such as the metabolomics and medications data in the UK Biobank. We proposed a novel modular transformer architecture, M-BioBERTa, with novel mechanisms and training protocols to cope with the block structure of the data. In addition, M-BioBERTa provides a simultaneous view of latent representations for the modalities and their balanced joint representation as well. We applied the approach to explore the stratification of depression and disease, medication, multimorbidity, and polypharmacy maps, demonstrating its applicability at scale using the UK Biobank. We also evaluated the approach in the quantitative prediction of disease and drug burden, which indicated clinically relevant improvements in predictive performance. We believe that the multimodal data fusion capabilities of M-BioBERTa hold promise not only for the subtyping of depression and its associated multimorbidities, but also for a broader range of health issues. By integrating diverse data types, M-BioBERTa can facilitate a deeper and more nuanced understanding of various health conditions. This enhanced understanding is essential to identify different health risk groups in different disorders, which could lead to better strategies for prevention and healthcare. Such applications of M-BioBERTa could be instrumental in personalizing treatment approaches and in the early identification of risk factors for a variety of diseases, thereby facilitating timely and targeted interventions.

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

# A APPENDIX

## A.1 TRAINING AND MODEL PARAMETERS

We used a uniform set of training parameters in all models to ensure consistency and comparability of the results. Any model parameters not explicitly listed in the tables below adhere to the default settings specified for RoBERTa. All models were trained using a Tesla V100 GPU and the training duration for each model was approximately one hour per epoch.

| Argument | Value |
|---|---|
| epochs | 6 |
| learning rate | 0.0003 |
| weight decay | 0.0 |
| adam beta1 | 0.9 |
| adam beta2 | 0.999 |
| warmup steps | 4000 |
| scheduler | linear decay |

Table 6: Training Arguments

| Parameter | M-BioBERTa | RoBERTa Encoder |
|---|---|---|
| encoder hidden layers | 2 | - |
| num hidden layers | 4 | 4 |
| hidden size | 312 | 312 |
| intermediate size | 1200 | 1200 |
| trainable parameters | 13,177,787 | 27,284,171 |
| number of parameters | 69,007,835 | 27,284,171 |

Table 7: Model Arguments for RoBERTa Encoder and M-BioBERTa

## A.2 DATA TOKENIZATION

Our tokenization process was applied across multiple data modalities to ensure consistent representation.

For diseases and prescriptions, we used specific coding systems. Diseases were represented using 3-character ICD10 codes, resulting in 1,127 distinct tokens. Prescriptions were tokenized using 4-character codes from ATC level 3, leading to 176 tokens.

The encoding strategy assigns two distinct tokens to each SNP, corresponding to the heterozygous genotypes and the homozygous minor allele genotypes (assuming biallelic SNPs). Importantly, in the case of the homozygous major allele genotype, no token is appended to the sequence. This approach yields a total of 1,400 SNP tokens.

Tokenization of metabolic and personal data involved converting these data into float sequences suitable for MixedTab embedding layers. Given the nature of float tokens, which do not inherently indicate their measurement, we introduced an additional token before each measurement token to specify its type. When combined with the tokens from the personal modality's categorical measurements, the disease tokens, and the prescription tokens, this resulted in 3,155 tokens for the tokenizer vocab.

To address the temporal component, dates were divided and tokenized into years, months, and days. Although months and days were straightforward, the year tokens only cover a range from 1934 to 2020. In particular, all the models used in this study shared the same vocabulary for uniformity.

To further demonstrate the tokenization process, we provide an example tokenization of a fictional patient. All values used in the example are for purely demonstration purposes. Consider a patient with the following information:

- List of Diseases: [F30, 2000-01-01]
- List of Prescriptions: [M01AE02, 2012-03-13, ], [M01AX05, 2019-06-29]
- Available Personal Information: [Sex Male], [Weight 56.3kg]
- Available Metabolomics Information: [Iron 171.3 mg/dl]
- Available SNPs: [rs2252755_G homozygous major allele], [rs2147377_A heterozygous]

The initial tokenization step outputs the following sequences, with all categorical information turned into tokens and standardized continuous values.

- Disease Tokens: <F30>
- Disease Year Tokens: <2000>
- Disease Month Tokens: <01>
- Disease Day Tokens: <01>
- Prescription Tokens: <M01A><M01A>
- Prescription Year Tokens: <2012><2019>
- Prescription Month Tokens: <03><06>
- Prescription Day Tokens: <13><29>
- Personal Tokens (Mixed float and tokens sequence): <Sex><Male><Weight>-0.13
- Metabolomics Tokens (Mixed float and tokens sequence): <Iron>1.3
- SNP tokens: <HETEROZYGOUS_rs2147377_A>

In the final step, special tokens ("beginning of sequence": <bos>and "end of sequence" <eos>) are added, and the tokens are switched to their token ids. The tokenizer has four separate vocabularies (token to id maps), year, month, day, and "information".:

- Disease Token IDs: [1, 301, 2]
- Disease Year Token IDs: [0, 67, 0]
- Disease Month Token IDs: [0, 1, 0]
- Disease Day Token IDs: [0, 1, 0]
- Prescription Token IDs: [1, 1257, 1257, 2]
- Prescription Year Token IDs: [0, 79, 86, 0]
- Prescription Month Token IDs: [0, 3, 7, 0]
- Prescription Day Token IDs: [0, 13, 29, 0]
- Personal Token IDs (Mixed float and tokens sequence): [1, 1325, 1359, 1335, -0.13, 2]
- Metabolomics Token IDs (Mixed float and tokens sequence): [1, 1556, 1.3, 2]
- SNP token IDs: [1, 2355, 2]

## A.3 THE MLM OBJECTIVE

In the Masked Language Model (MLM) pretraining phase, the model processes token sequences in which 15% of the non-special tokens are substituted with either a special <mask>token or another random token. The model then predicts logits for each token, with the categorical cross-entropy loss of the masked tokens serving as the loss function.

MLM is employed in two distinct phases of our methodology: first, in the pre-training of the modality encoders, and second, in the joint training of the entire architecture. During each encoder's pre-training, they are fed sequences with masked tokens and are trained using the MLM loss. In the joint training phase of the M-BioBerta architecture, only disease sequences are masked. Here, the encoders receive complete modality-specific sequences, and the MLM loss is computed exclusively at the decoder's output.

A.4    ABLATION STUDY

We first evaluate the effect of the proposed temporal embedding on the performance of the RoBERTa model in the coding of medical diagnoses, particularly in the prediction of International Classification of Diseases (ICD) codes. The modified model, RoBERTa + temporal, demonstrates improved accuracy over the baseline RoBERTa model across three metrics: code accuracy, block accuracy, and chapter accuracy. This improvement underscores the importance of temporal data in improving the predictive capabilities of natural language processing models within the medical domain. For detailed results, refer to the performance metrics outlined in Table A.4.

| Model | ICD Code Accuracy | ICD Block Accuracy | ICD Chapter Accuracy |
|---|---|---|---|
| RoBERTa | 20.06% | 25.47% | 36.15% |
| RoBERTa + temporal | **23.93%** | **29.31%** | **40.97%** |

Table 8: **How the temporal information affects model performance.**

We delve into the ablation study of the RoBERTa model enhanced with temporal embeddings (RoBERTa(t)), evaluating its performance in the context of Masked Sequence Modeling tasks related to medical diagnosis coding. The focus is on the addition of different single modality encoders: Prescriptions, Metabolomics, Personal, and Genetics. These modalities are integrated into the RoBERTa(t) model to assess their individual impact on the prediction of International Classification of Diseases (ICD) codes.

The results, as detailed in Table A.4, reveal varied impacts of these modalities on the performance of the model. The integration of Prescriptions data leads to a slight improvement in the three metrics: ICD Code Accuracy, ICD Block Accuracy and ICD Chapter Accuracy compared to the baseline RoBERTa(t) model. Similarly, the addition of Personal data results in the most notable improvement across all metrics.

In contrast, the integration of Metabolomics and Genetics data does not improve, in fact, marginally deteriorates the performance on most metrics compared to the baseline RoBERTa(t) model. These findings suggest that while certain data modalities like Prescriptions and Personal information can enhance the model's predictive accuracy, others like Metabolomics and Genetics may not contribute as effectively to the task of ICD code prediction in this specific modeling context.

| Model | ICD Code Accuracy | ICD Block Accuracy | ICD Chapter Accuracy |
|---|---|---|---|
| RoBERTa (temporal) | 23.93% | 29.31% | 40.97% |
| RoBERTa(t) + Prescriptions | 24.58% | 29.88% | 41.25% |
| RoBERTa(t) + Metabolomics | 23.77% | 29.11% | 40.66% |
| RoBERTa(t) + Personal | 24.93% | 30.34% | 41.51% |
| RoBERTa(t) + Genetics | 23.87% | 29.10% | 40.24% |

Table 9: **RoBERTa+temporal decoders, with a single modality encoder.** These metrics were used, to measure how well the model was able to perform that Masked Sequence Modelling tasks.

As detailed in Table A.4, investigates the effect of removing each of four specific modalities: Prescriptions, Metabolomics, Personal, and Genetics. This approach helps to understand the contribution of each modality to the model's overall performance.

The results show that the removal of Metabolomics and Genetics modalities leads to a notable increase in ICD Code Accuracy, ICD Block Accuracy, and ICD Chapter Accuracy, surpassing the performance of the complete fpM-BioBERTa model. This suggests that these modalities in the current settings might introduce noise or irrelevant information, thereby hindering the model's predictive accuracy.

On the contrary, removing prescriptions and personal data results in a slight decrease in performance across all metrics compared to the full fpM-BioBERTa model. This indicates that these modalities contribute positively to the model's ability to predict ICD codes.

| Model | ICD Code Accuracy | ICD Block Accuracy | ICD Chapter Accuracy |
|---|---|---|---|
| fpM-BioBERTa - Prescriptions | 24.57% | 30.07% | 41.29% |
| fpM-BioBERTa - Metabolomics | 25.51% | 30.93% | 42.30% |
| fpM-BioBERTa - Personal | 24.78% | 29.94% | 41.27% |
| fpM-BioBERTa - genetics | 25.76% | 31.1% | 42.47% |
| fpM-BioBERTa | 25.42% | 30.77% | 42.25% |

Table 10: **fpM-BioBERTa models, with a single modality removed.**

## A.5 EFFECT OF MODEL SIZE

In this section, we explore the effect of model size on the performance of the fpM-BioBERTa model. This study contrasts three versions of the model, differentiated by their number of layers: fpM-BioBERTa-small (2 layers), fpM-BioBERTa-base (4 layers) and fpM-BioBERTa-large (8 layers), with the aim of understanding how increasing model complexity impacts accuracy in ICD code prediction tasks.

As detailed in Table A.5, the results show a trend in which increasing the number of layers leads to marginal improvements in ICD Code Accuracy and ICD Block Accuracy. The fpM-BioBERTa-base model outperforms the small variant in both of these metrics. However, this upward trend does not continue linearly with the large model, as the fpM-BioBERTa-large model, although it has double the layers of the base model, shows only a slight increase in ICD Code Accuracy and no improvement in ICD Block Accuracy.

It should be noted that the accuracy of the ICD chapter does not consistently improve with increasing model size. The fpM-BioBERTa small model performs comparably to its larger counterparts in this metric, indicating that the additional complexity of larger models does not significantly enhance performance for chapter-level predictions.

This analysis suggests a nuanced relationship between model size and performance in multimodal biobank data processing tasks, particularly in Masked Sequence Modeling for ICD code prediction. It is important to consider that the benefits of larger models might be more noticeable with access to more data. The availability of larger datasets could potentially amplify the advantages of increased model complexity, leading to clearer distinctions in performance metrics across different model sizes.

| Model | Layers | ICD Code Accuracy | ICD Block Accuracy | ICD Chapter Accuracy |
|---|---|---|---|---|
| fpM-BioBERTa-small | 2 | 25.04% | 30.24% | 41.55% |
| fpM-BioBERTa-base | 4 | 25.42% | 30.77% | 42.25% |
| fpM-BioBERTa-large | 8 | 25.6% | 30.77% | 41.98% |

Table 11: **Model Performance Comparison Based on ICD Accuracy.** These metrics were used, to measure how well the model was able to perform that Masked Sequence Modelling tasks.

## A.6 FEATURE VECTOR CONSTRUCTION FOR REFERENCE MODELS

We prepared a special data set for the analysis with various machine learning methods by constructing a feature vector for each patient. This vector includes:

- **ICD10 and ATC Codes:** Each code in our data set is represented as a binary variable, indicating its presence or absence in a patient's record.
- **SNPs:** SNP use the same categorical encoding as for tokens.
- **Metabolomic and Personal Data:** Categorical data are converted into one-hot encoded format, while numerical data are standardized.

Imputation of Missing Data:

- **Categorical Variables:** Missing values in categorical data are imputed with -1, to differentiate them from other categories.
- **Continuous Variables:** For continuous variables, the missing values are imputed with the mean of the available data.

## A.7 FULL TEST METRIC TABLES FOR FINE-TUNE

| Model | Accuracy | Precision | Recall | F1 Score | AUROC | AUPRC |
|---|---|---|---|---|---|---|
| DT | 0.963 | 0.019 | 0.025 | 0.022 | 0.653 | 0.027 |
| LR | 0.910 | 0.027 | 0.127 | 0.044 | 0.658 | 0.028 |
| MLP | **0.968** | **0.085** | 0.101 | **0.092** | 0.666 | 0.044 |
| RF | 0.838 | 0.038 | 0.367 | 0.069 | 0.630 | 0.041 |
| XGB | 0.047 | 0.016 | **0.962** | 0.032 | 0.655 | 0.038 |
| RoBERTa | **0.968** | 0.018 | 0.023 | 0.018 | 0.713 | **0.276** |
| fpM-BioBERTa | 0.875 | 0.038 | 0.136 | 0.057 | **0.742** | 0.269 |

Table 12: F30-F39 Mood Related Disorders

| Model | Accuracy | Precision | Recall | F1 Score | AUROC | AUPRC |
|---|---|---|---|---|---|---|
| DT | 0.738 | 0.071 | **0.766** | 0.130 | 0.798 | 0.127 |
| LR | 0.924 | 0.140 | 0.379 | 0.204 | 0.801 | 0.147 |
| MLP | 0.502 | 0.042 | 0.839 | 0.079 | 0.782 | 0.114 |
| RF | 0.820 | 0.085 | 0.613 | 0.149 | 0.792 | 0.101 |
| XGB | **0.948** | **0.183** | 0.298 | **0.227** | **0.859** | 0.162 |
| RoBERTA | 0.867 | 0.092 | 0.282 | 0.129 | 0.756 | 0.394 |
| fpM-BioBERTa | 0.916 | 0.142 | 0.291 | 0.176 | 0.807 | **0.472** |

Table 13: E10-E14 Diabetes mellitus

| Model | Accuracy | Precision | Recall | F1 Score | AUROC | AUPRC |
|---|---|---|---|---|---|---|
| DT | 0.567 | 0.131 | 0.701 | 0.221 | 0.667 | 0.140 |
| LR | 0.666 | 0.137 | 0.529 | 0.217 | 0.657 | 0.138 |
| MLP | 0.408 | 0.116 | 0.871 | 0.205 | 0.681 | 0.145 |
| RF | 0.659 | 0.138 | 0.551 | 0.221 | 0.670 | 0.148 |
| XGB | 0.261 | 0.103 | **0.960** | 0.186 | 0.687 | 0.152 |
| RoBERTa | 0.672 | 0.145 | 0.540 | 0.216 | 0.660 | 0.303 |
| fpM-BioBERTa | **0.747** | **0.183** | 0.546 | **0.260** | **0.720** | **0.369** |

Table 14: I10-I15 Hypertensive diseases

| Model | MSE | RMSE | MAE | MAPE |
|---|---|---|---|---|
| DT | 1.10 | 1.05 | 0.651 | 96.1 |
| LiR | 1.11 | 1.06 | 0.657 | 98.5 |
| MLP | 1.08 | 1.04 | 0.624 | 89.4 |
| rf_reg | 1.08 | 1.04 | 0.649 | 95.8 |
| XGB | 1.08 | 1.04 | 0.646 | 95.6 |
| RoBERTa | **1.01** | 0.916 | 0.591 | 89.6 |
| pM-BioBERTa | **1.01** | **0.91** | **0.561** | **80.7** |

Table 15: Disease Burden

| Model | MSE | RMSE | MAE | MAPE |
|---|---|---|---|---|
| DT | 52.5 | 7.24 | 3.85 | 99.3 |
| LiR | 53.7 | 7.33 | 4.04 | 109.6 |
| MLP | 52.4 | 7.24 | 3.98 | 105 |
| rf_reg | 52.1 | 7.22 | 3.85 | 99 |
| XGB | 52 | 7.21 | 3.87 | 100 |
| RoBERTa | 46.6 | 6.33 | 3.23 | 96.2 |
| fpM-BioBERTa | **23.8** | **4.28** | **1.8** | **51.8** |

Table 16: Drug Burden

| Model | MSE | RMSE | MAE | MAPE |
|---|---|---|---|---|
| DT | 3.21 | 1.79 | 1.04 | 98.0 |
| LiR | 3.32 | 1.82 | 1.07 | 102.3 |
| MLP | 3.22 | 1.8 | 1.04 | 96.7 |
| rf_reg | 3.2 | 1.79 | 1.05 | 98.5 |
| XGB | 3.2 | 1.79 | 1.04 | 98.5 |
| RoBERTa | 2.18 | 1.41 | 0.793 | 80.2 |
| fpM-BioBERTa | **1.15** | **0.989** | **0.467** | **44.6** |

Table 17: Unique Drug Burden

# B    ADDITIONAL LATENT PLOTS

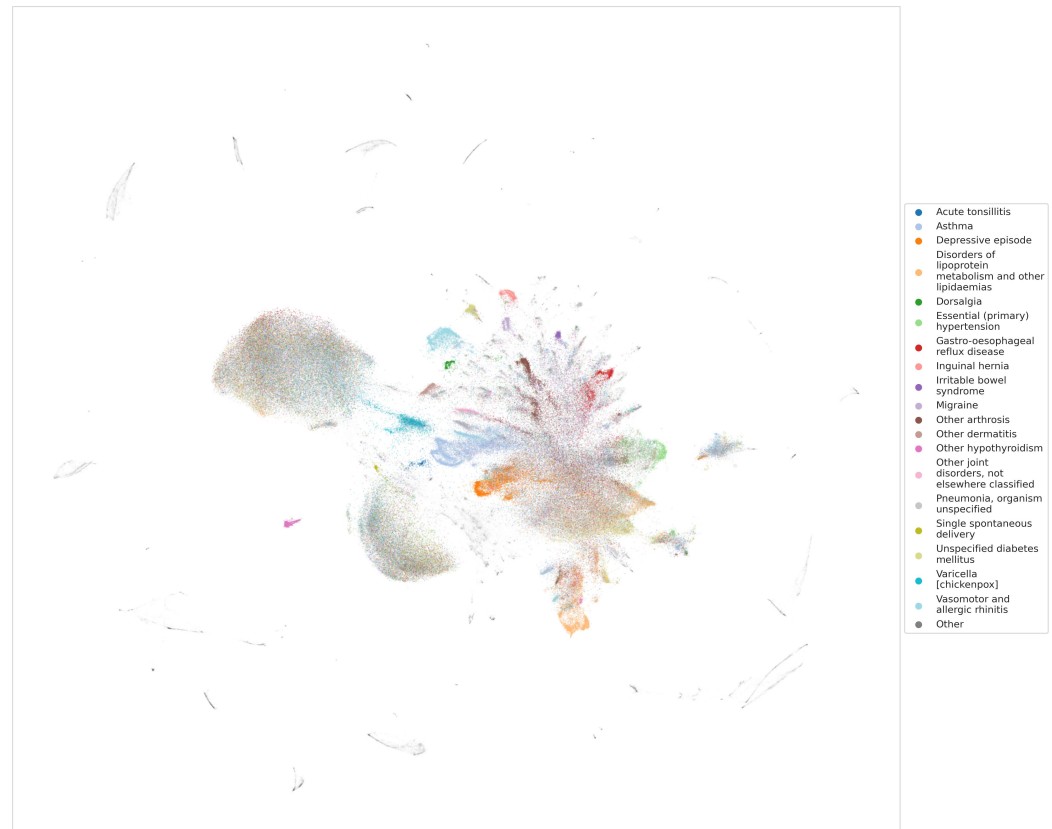

Figure 4: **Color coded by the first disease a patient had.**

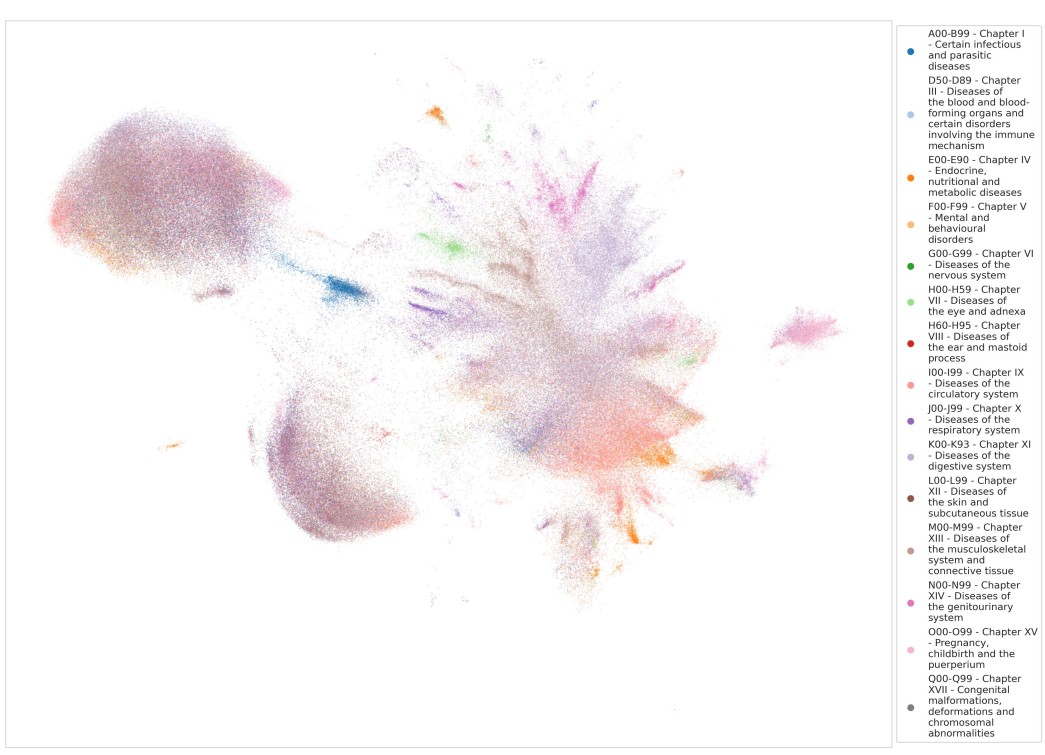

Figure 5: **Color coded by the most ICD Chapter from which a patient has the most diseases.**

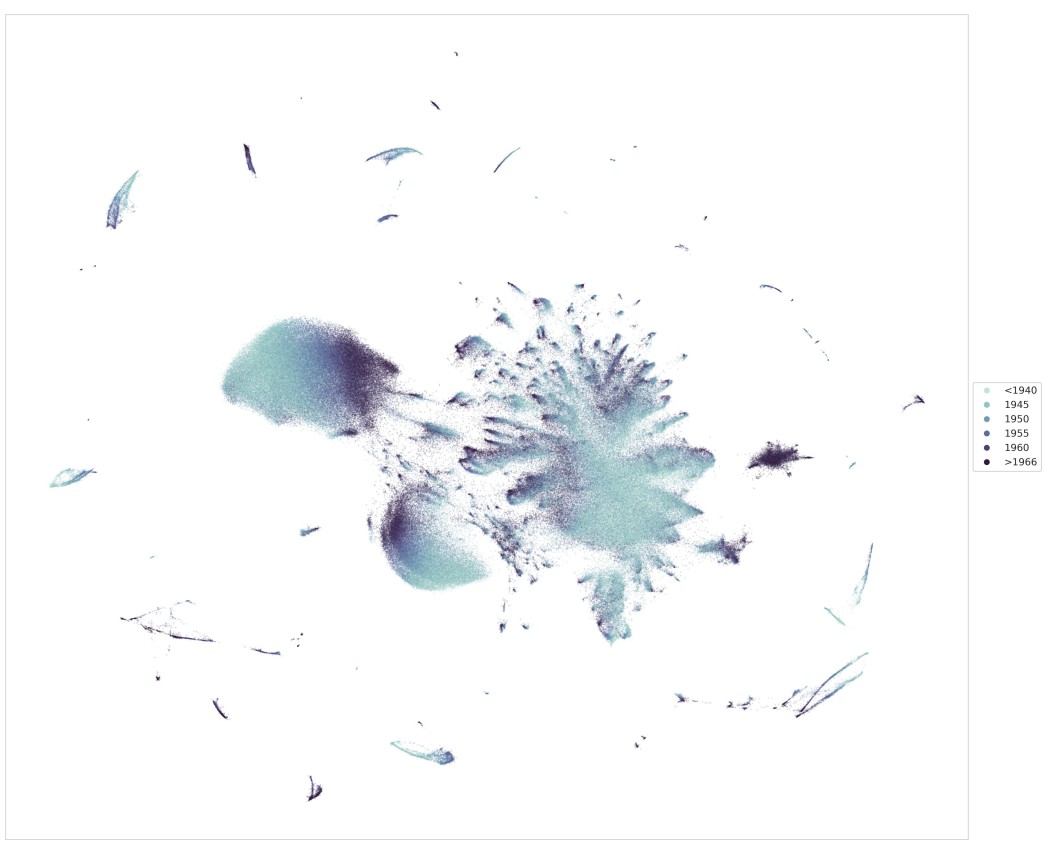

Figure 6: **Color coded by the birth year of each patient.**

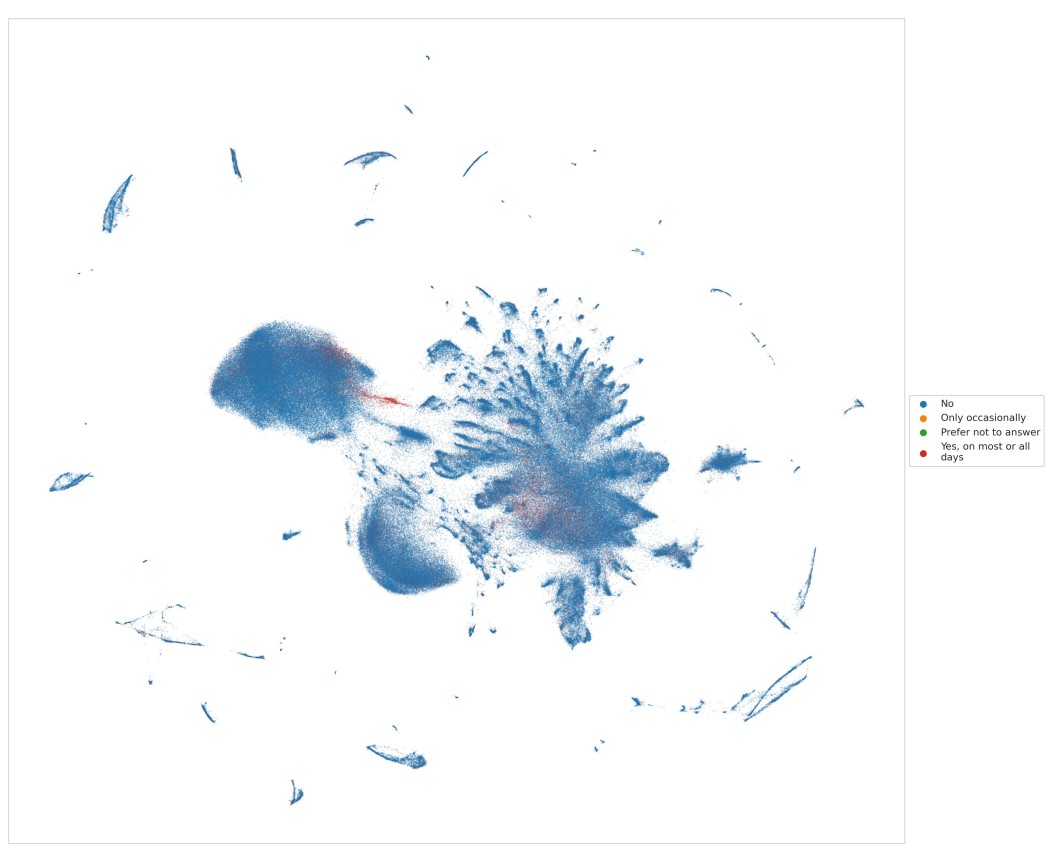

Figure 7: **Color coded by the patients smoking habits.**

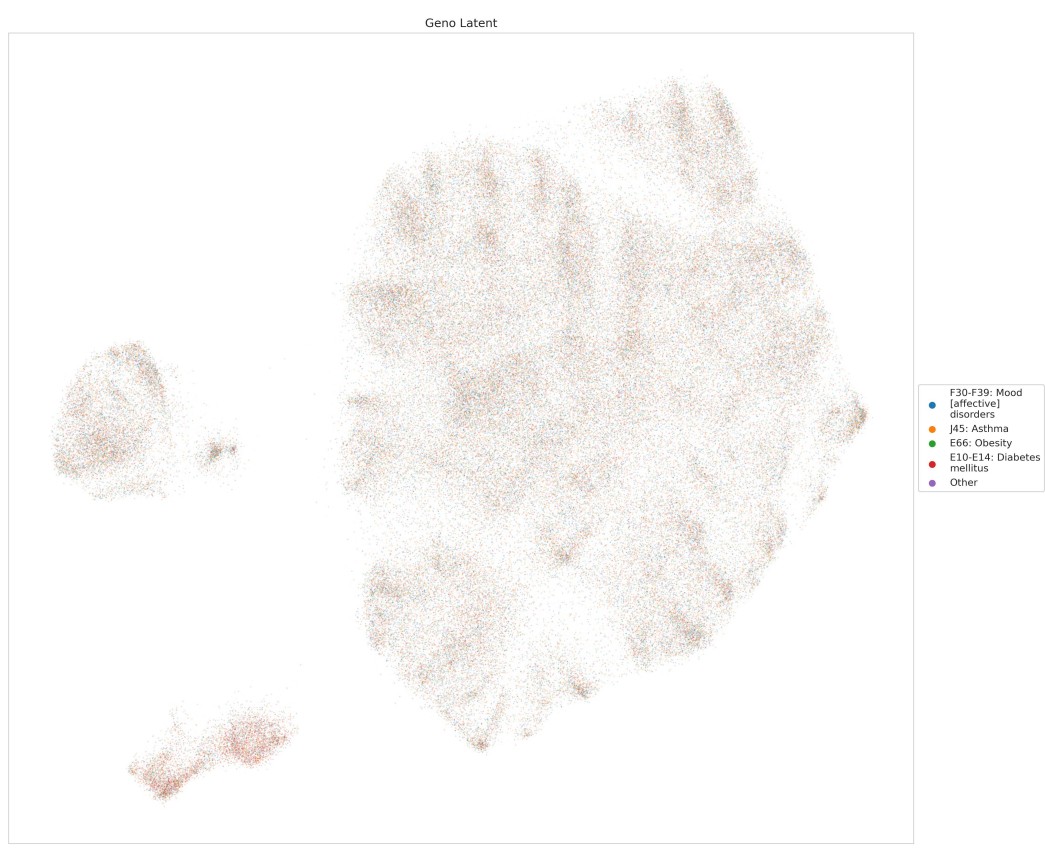

Figure 8: **The representation of the genetics encoder, color coded by the comorbidities of Major Depressive Disorder.** Though the plot shows little structure in this context, we find that it is able to model the provided genetic information well.

