# OpenReview forum: "M-BioBERTa: Modular RoBERTa-based Model for Biobank-scale Unified Representations"
_ICLR.cc/2024/Conference — Submitted to ICLR 2024_

### Official Review · Reviewer_e4Cc · 2023-10-23

**Soundness:** 3 good
**Presentation:** 1 poor
**Contribution:** 3 good
**Rating:** 3
**Confidence:** 5

**Summary:**

The research presents M-BioBERTa, a Transformer-based architecture designed to unify large-scale biobank data from diverse modalities and omic domains. The model's performance was evaluated using a wide array of data from the UK Biobank, which included genetic, demographic, laboratory, diagnostic, and drug prescription data, with a special emphasis on issues related to major depressive disorder such as multimorbidity and polypharmacy.

**Strengths:**

- Interesting idea on multi-modal learning over biobank data.
- Utilizes numerous modalities of a sample.
- Tackle the multi-modal nature through various forms of embeddings including temporal embedding.

**Weaknesses:**

- Presentation is unclear. Many important details are missing.
- The evaluation is unclear. Lots of experiments are needed to justify the model design.
- The approach is a bit ad hoc. Requires more justifications.

**Questions:**

Why Roberta model? It would be great to justify it.

How exactly is pre-training conducted? The authors mention that masked modeling is used. But here which part is masked? since there are multiple modalities, is it modality-specific masking? or others? have the authors experimented with any varieties of masking strategies?

"The genomics data, sourced from SNPs, was encoded to display only the minor and major allele in the sequence." what does this mean? every sample has both minor and major alleles? That does not make sense?

It would be great for the authors to give an exact example of what the input looks like.

Is there ablations on which modality is most useful? For example, is genetics useful at all since some works have shown that EHR information overrides with genetics signals.

Also, why are only 700 SNPs used? there are 800K genotyped SNPs, if not considering imputed arrays. These could be used for pre-training.

The authors used MMD disease as the prediction task. The data curation is also based on this task, which is limited.

There are many pre-trained foundation models for each modality, why not use them? but to train itself?

What are clusters in the Figure 2 and 3? There are little information with the current set of labels. Could the authors change to other more meaningful labels?

What is "Mean Absolute Percentage Error"?

Why do AUROC and AUPRC differ so much? For example in Table 4, E10-14, XGB has 0.859 the best AUROC but 0.162 worst AUPRC?

---

> ### Author Response · Authors · 2023-11-15
>
> Dear Reviewer e4Cc,
>
> Thank you for your valuable feedback, which has significantly contributed to the refinement of our paper.
>
> We have provided a revised version of the paper; for an overview of the changes, please refer to the general rebuttal.
>
>
> In addressing the points you raised:
>
> 1. and 2. We have tried to enhance the clarity of our paper by rephrasing critical sections. Additionally, we have included comprehensive explanations in the appendix, detailing pivotal aspects like tokenization, ablations, and feature extraction processes.
>
> 3. The addition of an "Ablations" section is intended to provide a more systematic evaluation. Our approach to managing different modalities is informed by contemporary studies on multi-modal transformers, as detailed in the referenced review ( Xu, Peng, Xiatian Zhu, and David A. Clifton. "Multimodal learning with transformers: A survey." IEEE Transactions on Pattern Analysis and Machine Intelligence (2023). ).  Furthermore, the applicability of BERT-like models for Electronic Health Records (EHR) data is supported by existing research such as Rasmy, Laila, et al. "Med-BERT: pretrained contextualized embeddings on large-scale structured electronic health records for disease prediction." NPJ digital medicine 4.1 (2021): 86.
>
>
>
> Addressing your queries:
>
> 1. We selected a BERT-based model due to existing research that has demonstrated its effectiveness. Investigating alternative transformer models as baselines could be a valuable direction for future research.
>
> 2. We have updated the appendix with a new section to elucidate the Masked Language Modeling (MLM) objectives, which were previously unclear. This section explains the dual application of the MLM objective: firstly, in pretraining the encoder for each modality with modality-specific token sequences, and secondly, in jointly training the four encoders with the decoder. In this latter phase, masking is exclusively applied to the decoder’s sequence, and correspondingly, the calculation of the MLM loss is confined to the decoder's output.
>
> 3. We have corrected the error concerning the SNPs in the tokenization section in our paper, thanks to your observation, and have updated the respective paragraph to ensure its accuracy.
>
> 4. Recognizing the unconventional nature of our tokenization approach, we have expanded the tokenization section in the appendix with a detailed example to help better comprehension.
>
> 5. An "Ablations" section has been added to the appendix, presenting an in-depth exploration of how key components influence model performance.
>
> 6. The choice of SNPs was based on their confirmed relation to Major Depressive: The choice of SNPs was motivated by their already established role to MDD, as our current model is not intended for SNP discovery at GWAS scale, such as linear mixed models, but to support cross-modal fusion and interpretation. We acknowledge the limitations imposed by computational constraints on the number of SNPs included. The ablations section offers insights into the minimal contribution of the genetic modality to overall performance. Future research could focus on better incorporating this genetic modality to enhance its effectiveness.
>
> 7. We apologize for any confusion and have clarified that only the genetics profile is specifically curated for MDD, not the broader UK BioBank data.
>
> 8. For the genetics encoder, exploring other pretrained foundation models could be beneficial. However, the uniqueness of other modalities to the UK BioBank data limits the applicability of foundation models trained on different datasets, such as MIMIC-VI.
>
> 9. We have updated the legends and captions of relevant figures to enhance their clarity and comprehensiveness.
>
> 10. We have revised the caption of 'Table 5: Performance Metrics for Regression Tasks' to include the definition of the MAPE acronym, standing for Mean Absolute Percentage Error. In our paper, we utilize MAPE as a standard metric for regression tasks. To briefly explain MAPE here, it measures the absolute error as a percentage of actual values.
>
> 11. The use of different metrics, such as AUPRC (Area Under the Precision Recall Curve), is particularly relevant in our case due to the heavily imbalanced nature of our datasets (as detailed in "Table 3: Statistics of Datasets for fine-tuning"). AUPRC is a more suitable metric for performance evaluation in such scenarios; see Saito, Takaya, and Marc Rehmsmeier. "The precision-recall plot is more informative than the ROC plot when evaluating binary classifiers on imbalanced datasets." PloS one 10.3 (2015): e0118432.
>
>
>
> We hope these revisions and explanations address your concerns and enhance the quality of our paper. Should you have any further suggestions or questions, please feel free to ask.

---

### Official Review · Reviewer_Xzvp · 2023-10-31

**Soundness:** 2 fair
**Presentation:** 3 good
**Contribution:** 3 good
**Rating:** 8
**Confidence:** 4

**Summary:**

This paper introduces M-BioBERTa, a modular transformer architecture for multimodal biobank data. M-BioBERTa ulitizes multiple unimodal RoBERTa encoders for individual data modalities, the model employs a unified RoBERTa decoder with cross-attention layers for cross modal training. Unimodal encoders are pre-trained on its respective modality before unified training. M-BioBERTa introduces novel elements like temporal embeddings for longitudinal data and mixed tabular embeddings for heterogeneous types. The authors utilize UK Biobank data to train and benchmark M-BioBERTa. M-BioBERTa outperforms baselines in disease and drug burden prediction, effectively handling missing modalities. Overall, it presents a novel approach for learning unified representations from multimodal biobank data.

**Strengths:**

1. Unimodal encoders enables handling of systematically missing multimodal data. Each encoder can be separately trained in parallel on their own data.  Cross modal attention fuses unimodal representations capturing interactions between modalities.

2. Temporal and tabular embeddings are vital for domain specific (biomedial) performance and M-BioBERTa pays clear attention to them

3. M-BioBERTa outperformed ML baselines on downstream predictive tasks using the UK Biobank data suggesting that it captures clinically relevant patterns.

4. Assuming typical transformer scaling laws M-BioBERTa is a great candidate to scale models using multimodal biomedical datasets.

**Weaknesses:**

1. There is a limited discussion on the scaling behavior of the proposed model architecture.

2. The paper does not include ablation studies to directly demonstrate the benefits of key components like the temporal embeddings.

3. Other transformer baselines are not considered for several downstream benchmarks.

**Questions:**

1. How sensitive is M-BioBERTa to the choice of pretraining objectives beyond MLM? Could other self-supervised tasks (T5, UL2) further improve the representations?

2. What is the impact of pretraining the encoders separately versus jointly pretraining the full model?

---

> ### Author Response · Authors · 2023-11-15
>
> Dear Reviewer Xzvp,
>
>
> We are grateful for your constructive feedback and insightful suggestions.
>
> We have provided a revised version of the paper; for an overview of the changes, please refer to the general rebuttal.
>
>
> Addressing the points you raised:
>
> 1. In response to your observation about model size and its impact on performance, we have incorporated an analysis in the appendix. This new section delves into the correlation between model size and final performance metrics, supplying a deeper understanding of our model's scalability and efficiency.
> 2. Recognizing the value of component analysis in our model, we have introduced an "Ablations" section in the appendix. This section examines the influence of various components, notably the temporal embedding and diverse modalities, on the overall results.
> 3. The primary objective of employing the RoBERTa baseline was to assess the impact of the proposed alterations on predictive efficacy. A further research direction of considerable interest involves examining the performance of alternative transformer models and investigating the implications of applying comparable alterations to these models.
>
>
> Responding to your inquiries:
>
> 1. Your question about the exploration of different pretraining objectives is indeed intriguing. While this aspect was not covered in our current research, we acknowledge its potential significance and consider it a promising avenue for future research.
> 2. We regret any confusion caused by the presentation of our models in the paper. To clarify, in "Table 2: Model Performance Comparison Based on ICD Accuracy," we compare three distinct models: "M-BioBERTa" (the jointly trained model), "pM-BioBERTa" (employing pretrained encoders), and "fpM-BioBERTa" (which freezes encoder weights, training only the decoder). We have revised the relevant section of the paper to elucidate this comparison more effectively, ensuring a clearer understanding of the distinct methodologies and their respective impacts on model performance.
>
>
>
> We hope these revisions and clarifications address your concerns and contribute to a more comprehensive understanding of our research. Should you have any further suggestions or questions, please feel free to ask.

---

> ### Comment · Reviewer_Xzvp · 2023-11-19
>
> Thanks for addressing my comments and updating the manuscript. I'm willing to raise my rating to 8.

---

### Official Review · Reviewer_aX8N · 2023-10-31

**Soundness:** 3 good
**Presentation:** 2 fair
**Contribution:** 2 fair
**Rating:** 5
**Confidence:** 4

**Summary:**

The authors presented a novel modular transformer based model to capture multi-modal EHR data and analyzed a large scale EHR dataset in the form of UK biobank.

**Strengths:**

The main strengths of the paper are as below
- The proposed idea is intuitive and makes a lot of sense when trying to model such large scale EHR datasets.
- The method shows performance improvement over selected baselines on both regression and classification tasks
- The appendix section drills down into the performance of the model around a number of key aspects such DM and Hypertension. In the context of health, it is often important for models to be investigated for their usage around specific usage criterion. The results are promising in these sections.

**Weaknesses:**

Some of the key aspects that can improve the model are as follows
- The main drawback is around the deleted baselines. Apart from XGB the selected baselines are rather weak. Even for the selected baselines, the feature selection and processing should be discussed in more details
- The method description is rather convoluted and difficult to follow. The main claim of the architecture is rather muddied and difficult to review for importance
- The modalities while discussed in the beginning is under-analyzed in terms of their contribution for predictive/modeling performance. The authors should consider ablation of modalities

**Questions:**

There are a couple of aspects that the authors can clarify
- For temporal embeddings, have the authors considered standards methods such as cosine embeddings?
- In section 3.4, while describing the unified cross-attention decoder, is the function of the "dedicated cross-attention units" to capture "inter-modal correlations" or intra-modal correlations?

---

> ### Author Response · Authors · 2023-11-15
>
> Dear Reviewer aX8N,
>
> Thank you for your insightful feedback and constructive critiques.
>
> We have provided a revised version of the paper; for an overview of the changes, please refer to the general rebuttal.
>
> In response to the concerns raised:
>
> 1. We appreciate your emphasis on the necessity for baseline models. Following your guidance, we have incorporated the missing baseline models pertinent to both regression and classification tasks. It's important to note that the inclusion of these models did not alter the previously reported best metrics results. We intentionally chose baseline models that are commonly used in healthcare settings, but we recognize the possibility of including additional models in subsequent studies. To enhance transparency, we have appended a detailed explanation of our feature selection method for these baseline models in the appendix.
>
> 2. To clarify our methodology, we have revised the relevant sections to ensure a more coherent and accessible presentation of our methods. We have also incorporated a concise statement of goals in the introduction for clearer understanding.
>
> 3. Acknowledging the importance of understanding the individual contributions of different modalities, we have included an "Ablations" section in the appendix. This section investigates the impact of each modality in two distinct scenarios: firstly, when a single modality supplements the disease sequences, and secondly, when one modality is excluded from the final fpM-BioBERTa architecture.
>
>
> Addressing your queries:
>
> 1. We regret any ambiguity in our first explanation about embedding methods and revised the relevant paragraph in the temporal embedding section. To clarify, our approach integrates all embedding methodologies of RoBERTa, including token type embedding and learned positional embedding, in conjunction with temporal embedding. We have opted for learned positional embedding instead of cosine embedding, aligning with current trends in the field.
>
> 2. In response to your request for greater clarity, we have revised the relevant paragraph for better comprehension. Additionally, we have updated Figure 1 to depict the information flow more accurately, specifically into the modality-specific cross-attention layers from the self-attention output.
>
> We hope these modifications and clarifications adequately address your concerns and enhance the comprehensibility of our work. Should you have any further suggestions or questions, please feel free to ask.

---

> > ### Comment · Reviewer_aX8N · 2023-11-22
> > **Thanks for the response**
> >
> > Thanks for the response. The additional results are helpful. However, considering the amount of updates I am of the opinion to keep my original review

---

> ### Author Response · Authors · 2023-11-22
> **Highlighting a key revision: deleted baselines**
>
> Dear Reviewer aX8N,
>
> Thank you for your original review and follow-up. We have addressed each of the weaknesses and queries you highlighted, without altering the core content and conclusions of our manuscript.  Importantly, we have incorporated the 'deleted baselines' you identified as the main drawback. In light of this, we respectfully request that you reconsider your final score.
>
> Best regards,\
> The Authors

---

### Author Response · Authors · 2023-11-15
**General rebuttal**

Thank you for the insightful comments; they help us improve our work significantly. In the general rebuttal, we address the updates made in our revision.


Correction in RoBERTa Baseline Results: We had to correct the results related to the RoBERTa baseline. The revised metrics, particularly in “Table 2: Model Performance Comparison Based on ICD Accuracy,” show a notable decrease in performance for the baseline. However, it is important to note that the change in the performance of downstream tasks is minimal. Despite these corrections, the overall conclusions drawn in our paper remain valid and unaffected.

We have made several updates to enhance clarity and readability:
- in the introductory paragraph of Section 3, M-BioBERTa,
- in subsection 3.4, detailing the Unified Cross-Attention Decoder,
- in Subsection 4.2, Models, we have included clearer descriptions of our models,
- in the evaluation subsection has been rephrased for improved understanding.

To address the need for more detailed information and transparency, we have expanded the appendix:
- An ablations section has been added to explore the effects of key components of our models.
- We included an example of tokenization to facilitate a better understanding of the process.
- The exact process of the Masked Language Model (MLM) pretraining phase is now thoroughly explained.
- A detailed description of the feature selection process for the reference models has been provided.

Alongside these major updates, we have also made several minor adjustments throughout the paper. These adjustments aim to enhance the overall clarity of the paper and correct minor errors.

In conclusion, we have revised the paper, reflecting on your feedback. The modifications do not alter the main claims of the paper and aim to provide a clearer, more accurate, and accessible account of our work.

---

### Author Response · Authors · 2023-11-20
**Request for further feedback**

Dear Reviewers,

We would like to kindly request your response to our rebuttal. We value your opinion on the new results and added evaluations, even if your assessment has not changed. Your feedback would be greatly appreciated!

We would also like to thank Reviewer Xzvp for responding to our rebuttal. We appreciate your time and effort in providing us with valuable feedback.

Best regards,
The Authors

---

### Meta-Review · Area_Chair_ehsw · 2023-12-10

**Metareview:**

The submission introduces M-BioBERTa, a modular transformer-based model designed for handling multi-modal Electronic Health Records (EHR) data, with a focus on large-scale datasets such as the UK Biobank. The model employs multiple unimodal RoBERTa encoders for individual data modalities and a unified RoBERTa decoder with cross-attention layers for cross-modal training. The paper also introduces new elements like temporal embeddings for longitudinal data and mixed tabular embeddings for heterogeneous data types.

The reviewers appreciate the intuitive nature of the model, its performance improvement over selected baselines, and the effective handling of systematically missing multimodal data. The use of temporal and tabular embeddings tailored for biomedical data is also seen as an interesting contribution.

However, there are notable concerns regarding the paper.  The paper's presentation is found to be lacking clarity by multiple reviewers. Important details are missing, and the method description is considered convoluted, making it difficult to understand the main claims and architecture. There is a consensus that the paper falls short in terms of evaluation. The choice of baselines is questioned, with a need for stronger and more relevant baselines. Additionally, the lack of ablation studies, particularly around key components like temporal embeddings, is noted. Reviewers have raised questions about the choice of the RoBERTa model, the specifics of pre-training, and the handling of genomics data. There is a call for more justification and clarity on these aspects. Concerns are raised about the model's scaling behavior and its generalizability beyond the specific tasks and datasets used.

The authors have responded to some of the concerns, providing additional results and clarifications. However, the response seems to have been insufficient in fully addressing all the key issues, particularly around the method description, evaluation rigor, and baseline selection.

Considering the strengths of the proposed approach in handling multi-modal EHR data and its potential impact in the biomedical domain, the paper presents potentially valuable contributions. However, the concerns regarding presentation clarity, evaluation depth, and methodological justifications are significant. These issues need to be addressed for the paper to meet the high standards of the conference.

**Justification For Why Not Higher Score:**

There are notable concerns regarding the paper.  The paper's presentation is found to be lacking clarity by multiple reviewers. Important details are missing, and the method description is considered convoluted, making it difficult to understand the main claims and architecture. There is a consensus that the paper falls short in terms of evaluation. The choice of baselines is questioned, with a need for stronger and more relevant baselines. Additionally, the lack of ablation studies, particularly around key components like temporal embeddings, is noted. Reviewers have raised questions about the choice of the RoBERTa model, the specifics of pre-training, and the handling of genomics data. There is a call for more justification and clarity on these aspects. Concerns are raised about the model's scaling behavior and its generalizability beyond the specific tasks and datasets used.

**Justification For Why Not Lower Score:**

N/A

---

### Decision · Program_Chairs · 2024-01-16

Reject